analytical chemistry

molecularly imprinted monolith,
capillary electrochromatography, sulfamerazine,
aquatic products

**Authors for correspondence:**
Shili Qin
e-mail: qinshili1103@163.com
Peng Wang
e-mail: pwang73@vip.sina.com

This article has been edited by the Royal Society of Chemistry, including the commissioning, peer review process and editorial aspects up to the point of acceptance.

# Determination of sulfamerazine in aquatic products by molecularly imprinted capillary electrochromatography

Shili Qin[1], Fenglong Jin[2], Lidi Gao[1], Liqiang Su[1], Yingjie Li[1], Shuang Han[1] and Peng Wang[3]

[1]College of Chemistry and Chemical Engineering, Qiqihar University, Qiqihar, Heilongjiang 161006, People's Republic of China
[2]Laboratory of Food testing, Qiqihar Food and Drug Administration, Bukui South Street 297, Longsha District, Qiqihar 161006, People's Republic of China
[3]State Key Laboratory of Urban Water Resource and Environment, School of Environment, Harbin Institute of Technology, Huanghe Road 73, Nangang District, Harbin 150000, People's Republic of China

SQ, 0000-0003-4853-803X

A molecularly imprinted monolith was prepared and evaluated for the special selective separation of sulfamerazine (SMR) by capillary electrochromatography (CEC). The single-step *in situ* polymerization method was applied through thermally immobilized vinyl groups of itaconic acid and a derivatization capillary column using SMR as the template. The monolith with optimal selectivity and permeability was performed at 45°C for 7 h according to the molar ratios of 1 : 4 : 10 (template/functional monomer/cross-linker). Under the optimized separation conditions of 75% acetonitrile in 20 mM phosphate buffer with pH 5.0, 15 kV applied voltage and 20°C column temperature, the imprinted monolith showed strong recognition ability for SMR and high column performance. Finally, the molecularly imprinted monolith coupled with the CEC method was successfully developed for the quantification of SMR in aquatic products, which was properly validated by a good linear relationship, recoveries and limit of detection. The coupling technique of the molecularly imprinted technology and CEC achieved pre-treatment enrichment and separation analysis in only one miniaturized chromatographic column.

# 1. Introduction

Sulfonamides (SAs) were among the first synthetic antibiotics and are widely used in humans and animals, but there is significant concern about their negative impacts on the ecosystem and human health after use [1,2]. It has been reported that China consumed 150 times more antibiotics than the UK, and more than 1000 tons of SAs were consumed by animals in 2013 [3]. Therefore, the efficient and accurate determination of trace SAs in some complex matrix samples is necessary for the testing organization. At present, chromatography and electrophoresis (CE) methods are mostly employed for the detection of SA residues, about 75% of the total [4]. Capillary electrochromatography (CEC) is considered to combine the advantages of the high separation efficiency of CE and the selectivity supplied by liquid chromatography [5,6]. It is important to realize that the CEC system operates with small sample quantities (nanolitre amounts) and sample vials used for the introduction of the sample only require microlitre quantities (typically greater than or equal to 50 μl), which is quite attractive for precious biological samples [7]. Several CEC studies on SA separations have been also reported by Wang & Ye [8] and Krivohlavek *et al*. [9]. In these cases, C18 packed columns were used to analyse SAs; however, long separation time or poor peak resolution was observed. The authors suggested using poly(divinylbenzene-octyl methacrylate) monolithic stationary phases for the successful detection of sulfamerazine (SMR) in meat samples using in-line solid phase extraction (SPE)-CEC with a mass spectrometry (MS) detector system. However, neither the MS detector system nor the off-line SPE for eliminating matrix interference were commercially available for the analysis of our experimental samples [10]. It is obvious that the development of a relevant stationary phase for the detection of SAs is necessary.

Molecularly imprinted technology is an easy and effective method for the preparation of tailor-made polymeric materials with molecular recognition abilities [11–14]. Compared with traditional studies of a molecularly imprinted polymer as the off-line or online SPE absorbent phase, a molecularly imprinted monolith coupled with a CEC system effectively combines pre-treatment enrichment and separation analysis in only one miniaturized chromatographic column, which has a unique selectivity towards the template [15–17]. Meanwhile molecularly imprinted polymer could improve the separation efficiency and overcome the low sensitivity of CEC [18]. Therefore, a CEC-based molecularly imprinted monolith has been increasingly used over a wide range of areas, such as studies on mechanisms, drug analysis, food analysis and chiral separation [19–22]. To the best of our knowledge, the practical application of a molecularly imprinted monolith coupled with CEC for SMR determination is not broad and has not been recently reviewed.

In this study, SMR and itaconic acid were selected as the template and the functional monomer, respectively. Then, a single-step *in situ* polymerization method was applied inside a 100 μm id derivatization silica capillary by 3-trimethoxysilyl-propyl methacrylate (γ-MAPS). Moreover, the reaction conditions (molar ratios, temperature and reaction time) of the molecularly imprinted monolith and the separation conditions (separation voltage, organic modifier concentration, pH value of the buffer, salt concentration of the buffer and column temperature) of CEC were optimized by separating SMR and its analogue, sulfamethazine. Also, the imprint effect of the prepared monolith was explored by using seven different kinds of SAs. Finally, when the imprinted monolith coupled with the CEC method was established and applied to the determination of SMR in aquatic products by adding standards to the blank, a satisfactory result was obtained.

# 2. Material and methods

## 2.1. Reagents and chemicals

SMR, sulfathiazole, sulfadiazine, sulfadimethoxine, sulfisoxazole, sulfamethoxazole, sulfamethazine, ethyleneglycol dimethacrylate (EGDMA), γ-MAPS and itaconic acid were purchased from Sigma-Aldrich (Steinheim, Germany). N,N-dimethylformamide (DMF), 2,2'-azobis (2-isobutyronitrile) (AIBN), cyclohexanol, dodecanol, acetic acid and sodium phosphate were purchased from J&K Chemical (Beijing, China). High-performance liquid chromatography (HPLC)-grade methanol was obtained from Kermel (Tianjin, China). All other reagents were of analytical grade. Phosphate buffer was used for the pH range 3.0–7.0, and mixed with acetonitrile to form the different proportions of the mobile phase. The analyte mixture was prepared in acetonitrile and filtered through a 0.45 μm membrane. The concentration of the analytes was 1 mg ml$^{-1}$.

**Figure 1.** Preparation procedures of the molecularly imprinted monolith.

## 2.2. Apparatus

All of the CEC experiments were performed on Agilent HD3D CE instruments (Hewlett–Packard, Waldbronn, Germany), containing a diode array detector and ChemStation software for data processing. Fused silica capillaries with dimensions of 100 μm id and 375 μm od were purchased from the Yongnian Optic Fibre Plant (Hebei, China). Scanning electron microscope (SEM) images of the monolith and Fourier transform infrared (FT-IR) spectra of the monolithic materials were obtained using an S-4300 SEM (Hitachi Limited, Japan) and a AS380 spectroscope (Nicolet, America), respectively.

## 2.3. Monolith preparation

A fused-silica capillary was first pre-treated as follows: the capillary was rinsed with 1.0 M NaOH for 1 h then with water for an additional 10 min; next, it was flushed with 1.0 M HCl for 30 min and water to a neutral pH; finally, it was rinsed with acetone for 10 min to achieve a fast-drying column. The capillary was dried under a nitrogen gas flow overnight at room temperature.

The procedures for the preparation of the imprinted monolith are shown in figure 1. Derivatization column: the γ-MAPS/acetone (1 : 1, v/v) mixture was used to fill the first pre-treated capillary and was maintained for 24 h at 40°C in a water bath. Then, the capillary was flushed with methanol and dried under a nitrogen flow. Molecularly imprinted monolith: the different molar ratios of template (30 μM)/functional monomer/cross-linker, AIBN (9.7 μM) as an initiator, cyclohexanol and dodecanol as porogenic agents were all dissolved in 150 μl DMF as the solvent. Then the uniformly pre-polymerized mixture was deoxidized by nitrogen for 15 min and introduced into the derivatization column. The ends of the capillary were sealed with soft plastic rubber and submerged in different temperature water baths. The resultant monolith was washed with methanol/acetic acid (9 : 1, v/v) by

an HPLC pump to remove the template and the residual reagents and was then washed with pure methanol overnight. The non-imprinted monolith was prepared and treated using the same method as the imprinted monolith.

## 2.4. CEC method

The capillaries were initially flushed for 10 min with the required mobile phase and then equilibrated by applying a voltage of 20 kV until a stable current and baseline were achieved. Between the sample runs, the columns were rinsed with methanol, pure water and buffer for 5 min intervals. The total length of the capillary was 355 mm, and the effective length was 255 mm. The pressure sample injection method was 30 mbar for 10 s. The samples were monitored at 275 nm.

## 2.5. Method validation and application

The method employed was validated before the practical sample application. An external standard method was used for quantitative analysis. A series of working standards of SMR were diluted with ultrapure water to seven concentrations ranging from 0.1 to 20 mg kg$^{-1}$. Precision and recovery experiments were performed by analysing a blank aquatic sample spiked with a standard solution five consecutive times at 0.5, 2 and 20 mg kg$^{-1}$. Additionally, the limit of detection (LOD) was calculated using the noise of the CEC profile.

Certain aquatic products (fish and shrimp) were purchased from the local market and used as test samples in the study. The samples (5.0 g) were weighed precisely, vortex-mixed and then ultrasonicated three times with 5.0 ml acetonitrile for 10 min. Then, the mixture was centrifuged for 6 min at 10 000 r.p.m. The clear supernatant was collected and evaporated to dryness under a stream of nitrogen. The residue was re-dissolved in 1.0 ml of the buffer, pH 5.0, and filtered through a 0.45 μm pore-size membrane for CEC analysis.

# 3. Results

## 3.1. Preparation of the molecularly imprinted monolith

Concerning the reaction format, the single-step synthetic route was applied, which was not only simple and fast but also generated super-porosity for the selectivity of monolith [23]. Therefore, the optimization factors of the ratio of the template/functional monomer/cross-linker, the reaction time and the reaction temperature were investigated in the preparation of the molecularly imprinted monolith [24,25].

The resultant column performance was evaluated by the permeability and the ratio of the retention factor ($I$) in which $I = K_1/K_2$, where $K_1$ and $K_2$ are the retention of SMR on the imprinted monolith and non-imprinted monolith, respectively, and $K = (t_R - t_0)/t_0$, where $t_R$ and $t_0$ are the migration time of the retained peak and non-retained neutral thiourea, respectively. Meanwhile to illustrate the permeability of the imprinted monolith, a process was established so that, when the pressure was set to 10 MPa, pure methanol could flow out. Also, the flow rate was faster than 30 μl min$^{-1}$, 20–30 μl min$^{-1}$ and lower than 20 μl min$^{-1}$, indicating that the permeability of the prepared monolith was better, good and low, respectively. The results are shown in table 1.

### 3.1.1. Optimized ratios of reactants

In the preparation of molecularly imprinted monolith, itaconic acid as the functional monomer was responsible for the binding interaction in the imprinted binding sites [26,27]. Itaconic acid with two carboxylate groups was used as a proton donor and a hydrogen bonder. In this study, both hydrogen bonds (carboxylate of itaconic acid and amino of SMR) and electrostatic interactions (acid carboxylate of itaconic acid and basic pyrimidine of SMR) could be formed between the template and the functional monomer simultaneously. Compared with the molecularly imprinted polymer using methacrylic acid or 4-vinylpyridine as the functional monomer, the imprinted polymer using itaconic acid with higher charged functionalities presented better separation efficiency and recognition ability [28]. As shown in table 1, when the ratio of the template/functional monomer was lower than 1 : 4 (M1), the $I$-value (1.87) was lower. The reason for this may be that the weaker interaction between the template (SMR) and itaconic acid resulted in a limited number of affinity sites in the monolith to recognize the target analytes. However, a higher dosage of itaconic acid could not achieve a better value of $I$ (3.34) than that of M2

**Table 1.** Preparation protocol for molecularly imprinted monolith.

| imprinted monolith | molar ratio[a] | per cent of porogenic agent (%) | temperature (°C) | time (h) | permeability | I |
|---|---|---|---|---|---|---|
| M1 | 1 : 2 : 10 | 12 | 45 | 7 | good | 1.87 |
| M2 | 1 : 4 : 10 | 12 | 45 | 7 | good | 3.34 |
| M3 | 1 : 6 : 10 | 12 | 45 | 7 | low | 1.22 |
| M4 | 1 : 4 : 8 | 12 | 45 | 7 | better | 2.19 |
| M5 | 1 : 4 : 14 | 12 | 45 | 7 | no | —[b] |
| M6 | 1 : 4 : 10 | 4 | 45 | 7 | no | — |
| M7 | 1 : 4 : 10 | 8 | 45 | 7 | low | 1.14 |
| M8 | 1 : 4 : 10 | 20 | 45 | 7 | — | — |
| M9 | 1 : 4 : 10 | 12 | 50 | 7 | good | 2.01 |
| M10 | 1 : 4 : 10 | 12 | 55 | 7 | no | — |
| M11 | 1 : 4 : 10 | 12 | 45 | 6 | — | — |
| M12 | 1 : 4 : 10 | 12 | 45 | 8 | low | 1.06 |
| M13 | 1 : 4 : 10 | 12 | 45 | 9 | no | — |

[a]The molar ratio of template (30 μM)/functional monomer/cross-linker.
[b]Non-resolved.

because the self-polymerization or copolymerization of the functional monomer reduced the selective recognition abilities, and the permeability decreased. EGDMA as the cross-linker provided a higher mechanical strength and greater rigidity of the molecularly imprinted monolith, the tests of cross-linker ratios were discussed. The results of three different ratios (M2, M4 and M5), shown in table 1, demonstrated that the permeability of the monolith decreased with an increasing amount of EGDMA from 8 mmol to 14 mmol. It was clear that, with a higher amount of cross-linker, the permeability decreased, as reflected in the enhanced polymerization reaction. It was not possible to further evaluate the column in the CEC mode. Although the permeability of M4 was the best, the selective ability was weaker than that in M2. It could be that, when the amount of cross-linker was reduced, the synthetic polymer was too soft to firmly bind on the inter-surface of the derivatization monolith and form stable affinity sites; therefore, the molecular recognition ability of the imprinted monolith weakened. Meanwhile, the optimization of the porogenic agent amount could be controlled, which directly affected the permeability of the molecularly imprinted monolith. Different percentages of dodecanol and cyclohexanol were selected as binary porogenic agents for the preparation of the imprinted monoliths (M2, M6–M8). As can be seen in table 1, when the percentage of porogenic agent was 4% a dense monolith was obtained, preventing its use in CEC mode; however, too much porogenic agent made polymers have a soft gel-like appearance. Although both synthesized monoliths presented relatively good permeability when the percentage of porogenic agent was 8% or 12%, the latter had better special recognition for SMR. Thus, with the other reaction conditions being equal, M2 was selected as the optimal reaction ratio.

### 3.1.2. Optimized reaction temperature and reaction time

The temperature and reaction time were also important for the immobilization of the vinyl group for polymerization, particularly for a micro-reaction environment such as the interior of the capillary column [29]. The initiation temperature of AIBN was higher than 40°C, and the prepared monolith was impermeable at 55°C according to the experimental results. The permeability of the imprinted monolith was good at 45°C and 50°C, but the recognition ability was better at 45°C. The result revealed that mild reaction conditions were preferred to acquire better long-term stability. Meanwhile a longer or shorter reaction time could cause a high degree of densification or easy collapse of the imprinted monolith. Therefore, the optimum temperature and reaction time were 45°C and 7 h.

In summary, the molecularly imprinted monolith with good permeability and recognition ability was prepared under the reaction conditions of a 1 : 4 : 10 ratio of template/functional monomer/ cross-linker, a percentage of porogenic agent of 12%, a reaction temperature of 45°C and a reaction time of 7 h.

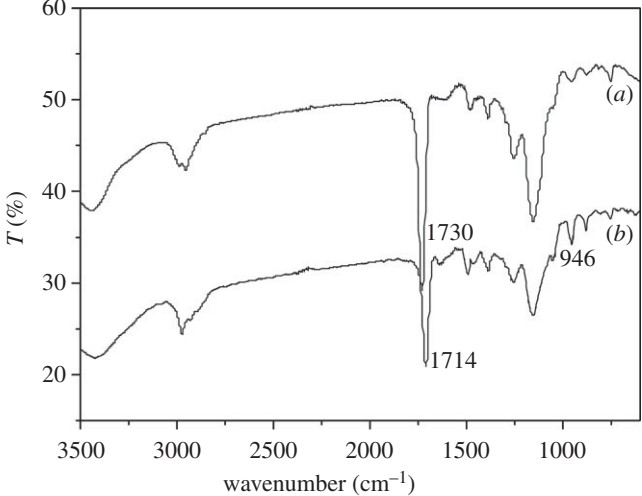

**Figure 2.** Infrared spectroscopy of bare monolith (*a*) and molecularly imprinted monolith (*b*).

## 3.2. Characterization of molecularly imprinted monolith

### 3.2.1. Infrared spectroscopy

The bare monolithic material and the imprinted monolithic material were characterized by IR, as shown in figure 2. The spectrum lines (*a*) and (*b*) were observed for the stretching vibration of −OH (3424 cm$^{-1}$) and the antisymmetric and symmetric stretching vibration of saturated C−H (2974 and 2928 cm$^{-1}$). Compared with spectrum line (*a*), spectrum line (*b*) showed that the absorption peak of the carbonyl group shifted to lower wavenumbers (1714 cm$^{-1}$) under the influence of the conjugate action because of the introduction of itaconic acid. Meanwhile, the disappearance of stretching vibration of C=C (1629 cm$^{-1}$) and the out-of-plane bending vibration of −OH (946 cm$^{-1}$) proved that itaconic acid was successfully immobilized.

### 3.2.2. Structure of the monolithic stationary phase

Visualization of the microstructure of the imprinted monolith was performed through SEM on several capillary cross sections. Figure 3 shows that many macro-pores and flow-through channels formed in a dense network skeleton of the imprinted polymer, which offered a high external porosity, a high permeability and low column hydraulic resistance. The imprinted polymer was closely linked with the inner surface of the capillary, which illustrated that the covalent attachment points between the polymer and the derivatization capillary wall ensured definite stability and strength of the stationary phase.

## 3.3. CEC separation of SMR and sulfamethazine

In this study, the imprinted selectivity of the prepared monolith was examined by the separation of SMR and its structural analogue (sulfamethazine). The separation conditions, including the pH value of the buffer, organic solvent, salt concentration of the buffer and the separation temperature, were systematically investigated for imprint selectivity. The resolution was evaluated to understand the recognition mechanism on the imprinted monolith and was calculated from the equation $R_s = 2(t_2 - t_1)/(w_1 + w_2)$, where $t_1$ and $t_2$ are the retention times of the first and second eluted compounds, respectively, and $w_1$ and $w_2$ are the baseline peak widths of the first and second eluted compounds, respectively.

### 3.3.1. Effect of the pH value

The pH value of the mobile phase played an important role in separation in the CEC analysis, as it altered the charges of the analytes and the polymer; thereby, the retention time, $R_s$ and electroosmotic flow (EOF) of the monolith were affected [30]. In this study, according to the p$K_a$ values of itaconic acid (3.85 and 5.45), SMR (2.17 and 6.77) and sulfamethazine (2.28 and 7.42), the different pH values

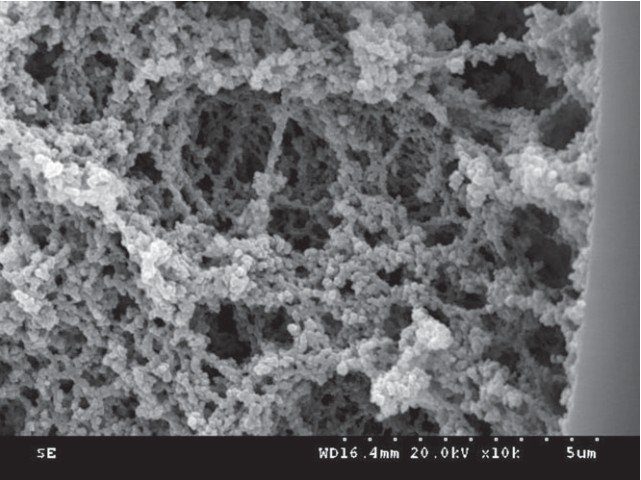

**Figure 3.** Scanning electron micrograph of the molecularly imprinted monolith.

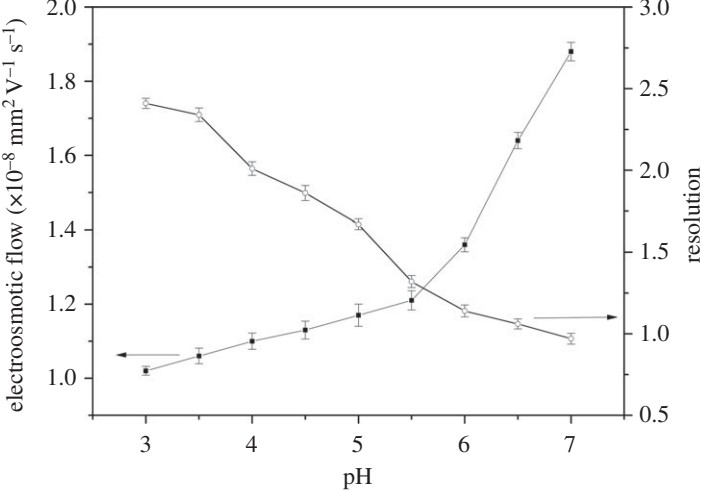

**Figure 4.** The effect of pH buffer solution on $R_s$ and EOF.

ranging from 3.0 to 7.0 were investigated. As shown in figure 4, as the pH value increased, the EOF mobility increased from $1.02 \times 10^{-8}$ to $1.88 \times 10^{-8}\,\mathrm{m^2\,V^{-1}\,s^{-1}}$. This could be attributed to the greater effective charge origination from the highly dissociated carboxylate groups of the excess functional monomer. Meanwhile the pH value was in range 3.0–5.0 and $R_s$ decreased from 2.41 to 1.67. Based on the Henderson–Hasselbalch equation for acids and bases, when the pH value of the buffer was 5.0, the net charges of SMR and sulfamethazine were $-0.02$ and 0.00, respectively [31]. Moreover at this pH value, the functional monomers partly dissociated, which not only generated the EOF but also made the hydrogen bonding between the analytes and the functional groups stronger. However, when the pH value was higher than 5.0, the function monomer dissociated completely and lost the affinity sites for the template molecule so that SMR and sulfamethazine could not reach the baseline separation ($R_s = 1.32$–0.97). In consideration of the high efficiency of CEC and the selectivity of the imprinted monolith, pH 5.0 was selected as the optimal condition for the separation in the following experiments.

### 3.3.2. Effect of organic solvent

The addition of organic solvent altered the polarity and viscosity of the buffer solution [32]. The effect of acetonitrile content on EOF and $R_s$ was investigated over a range of 60–85% (v/v) and in 20 mM phosphate (pH 5.0), as shown in figure 5. The EOF velocity was influenced by acetonitrile content

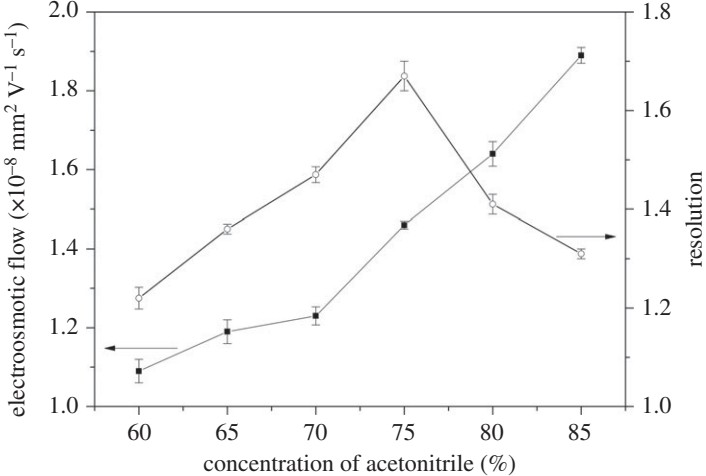

**Figure 5.** The effect of organic solvent content on $R_s$ and EOF.

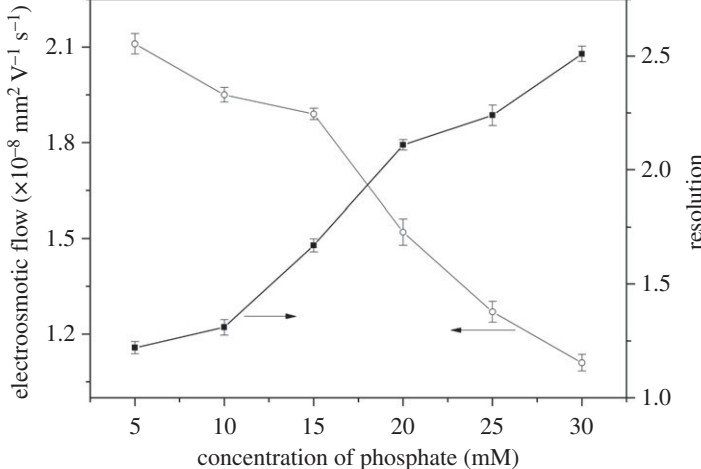

**Figure 6.** The effect of phosphate concentration on $R_s$ and EOF.

through the ratio of permittivity to viscosity, $\varepsilon_r/\eta$. With an increasing acetonitrile content, the $\varepsilon_r/\eta$ increased, the ionic strength of the mobile phase decreased and the zeta potential increased; thus, EOF increased. $R_s$ increased with the increase of the acetonitrile content from 60% to 75%. This result implied that, when the organic solvent content was higher, the hydrogen bonding interaction between SMR and the imprinted monolith in the hydrophobic environment was stronger. Therefore, the affinity sites increased, and the ability for specific recognition was stronger. However, when the acetonitrile content changed from 80% to 85%, recognition decreased because, with the increase of EOF, the retention time was shorter, and the analytes did not interact well with the imprinted monolith, resulting in poorer recognition and selectivity. When the acetonitrile content was less than 60%, the EOF was slow, and the band-broadening increased. When the acetonitrile content was higher than 85%, bubbles were easily formed. In terms of $R_s$, 75% acetonitrile was the final chosen content.

### 3.3.3. Effect of salt concentration

In this work, a low concentration of phosphate was applied and the effects on EOF and $R_s$ were studied by changing the concentration from 5 to 30 mM (pH 5.0) with 75% acetonitrile. Figure 6 shows that EOF decreased by increasing the salt concentration. This result was because the high ionic strength resulted in a relatively thinner double layer, increasing the chance of forming ion pairs and decreasing the effective charges. The $R_s$ of SMR and sulfamethazine increased with increasing salt concentration. The decrease of EOF indicated a longer retention time, improving the recognition. When the salt concentration was too high (greater than 30 mM), Joule heating was caused. However, when the salt concentration was

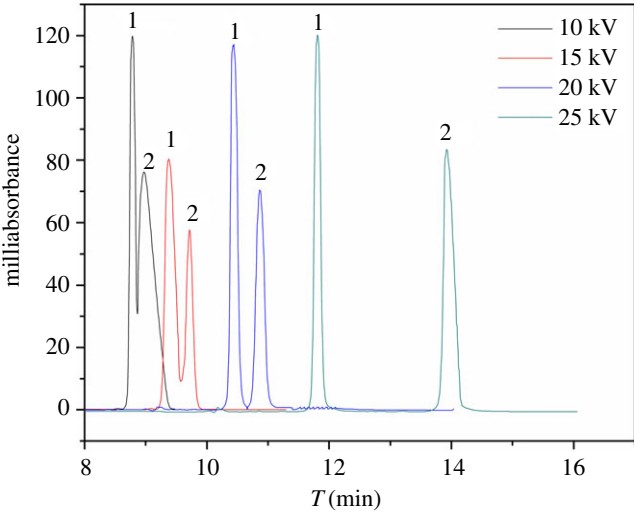

**Figure 7.** Effect of applied voltage. (*a*) 10 kV, (*b*) 15 kV, (*c*) 20 kV, (*d*) 25 kV. 1, sulfamethazine; 2, SMR.

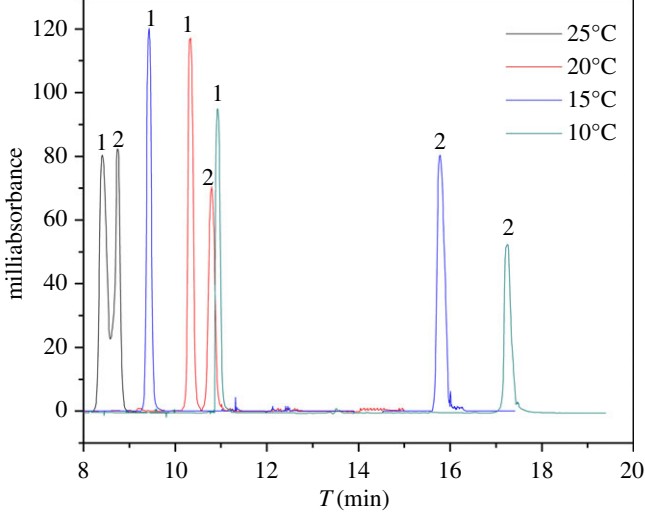

**Figure 8.** Effect of column temperature. (*a*) 10°C, (*b*) 15°C, (*c*) 20°C, (*d*) 25°C. 1, sulfamethazine; 2, SMR.

at 5 mM, the baseline separation of SMR and sulfamethazine was not obtained. Considering these results, the optimum salt concentration was 20 mM.

### 3.3.4. Effect of applied voltage and column temperature

In this study, the applied voltage and column temperature that could fine-tune $R_s$ were investigated by varying their values from 10 to 25 kV and from 10°C to 25°C, respectively, using the mobile phase containing 75% acetonitrile in 20 mM phosphate buffer at pH 5.0. The results shown in figures 7 and 8 indicate that, with an increasing applied voltage and column temperature, EOF increased and the $R_s$ decreased because of the shorter retention time causing weaker recognition ability. Also, an excessive voltage and temperature would lead to deterioration of the separation by the Joule heat effect. Briefly, an optimized applied voltage and column temperature of 15 kV and 20°C, respectively, were chosen.

In conclusion, the optimum chromatographic separation conditions were 75% acetonitrile in 20 mM phosphate buffer with pH 5.0, 15 kV applied voltage and 20°C column temperature.

## 3.4. Imprinted effect evaluation of molecularly imprinted monolith

To test the function of the molecular recognition of the imprinted monolith, the separations of seven SAs were chosen for the selectivity study. Among them, sulfisoxazole, sulfamethazine and sulfathiazole

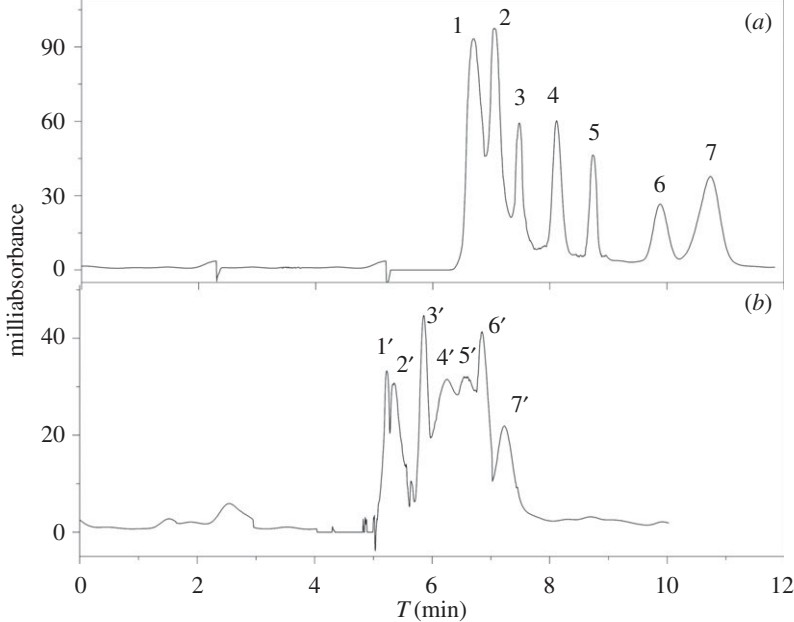

**Figure 9.** Electrochromatogram of molecularly imprinted monolith (*a*) and non-imprinted monolith (*b*) for separations of seven sulfonamides. 1, sulfadimethoxine; 2, sulfisoxazole; 3, sulfamethoxazole; 4, sulfathiazole; 5, sulfadiazine; 6, sulfamethazine; 7, SMR; 1′, sulfadimethoxine; 2′, sulfisoxazole; 3′, sulfamethazine; 4′ SMR; 5′, sulfadiazine; 6′, sulfamethoxazole; 7′, sulfathiazole.

**Table 2.** Comparison of $K$- and $I$-values of seven SAs on molecularly imprinted monolith and non-imprinted monolith.

| analyte | $K_1$ | $K_2$ | $I$ |
|---|---|---|---|
| SMR | 2.71 | 0.81 | 3.34 |
| sulfamethazine | 2.43 | 0.84 | 2.89 |
| sulfadiazine | 2.08 | 1.11 | 1.87 |
| sulfathiazole | 1.89 | 1.24 | 1.52 |
| sulfamethoxazole | 1.69 | 1.16 | 1.46 |
| sulfisoxazole | 1.55 | 1.16 | 1.34 |
| sulfadimethoxine | 1.43 | 1.13 | 1.26 |

belonged to short-acting SAs; sulfamerazine, sulfadiazine and sulfamethoxazole belonged to medium-acting SAs; sulfadimethoxine belonged to long-acting SAs based on the pharmacological properties and clinical uses [33]. The mixtures with similar structures, properties and drug effects could offer a good way to investigate the recognition mechanism. The $K$, $I$, $R_s$ and chromatography of each SA on the imprinted monolith and non-imprinted monolith were calculated and are shown in table 2 and figure 9. These results illustrated four points. (i) The $I$-values and the retention time of SMR were the highest and longest on the imprinted monolith; in other words, it showed better recognization ability for SMR. The reason for this is that the formation of the recognition cavities in the imprinted monolith was primarily derived from the shape and size of the template molecular and three-dimensional network formed by the functional monomers. (ii) Under the same separation conditions, there were the less differences in the $K$ values and the mixture was not separated on the non-imprinted monolith. The main reason for this was the desultory polymerization of the functional monomers, which could not producing site-specific and site-identifiable cavities. (iii) The more dissimilar the structure was to SMR, the earlier the peak, and the smaller the $I$. It may be that the amino groups of seven SAs in similar positions could form hydrogen bonds with the carbonyl of itaconic acid, but the different heterocyclic rings participated in different interactions with itaconic acid. In addition, the different molecular size and spatial structure also determined the special recognition of the imprinted cavities. It can be deduced from the structures of the SAs that their effect

**Table 3.** Reproducibility of the imprinted monolith.

| type and numbers ($n = 7$) of experiment | $R_s$ (% RSD) | $t_{SMR}$ (% RSD) | $t_{sulfamethazine}$ (% RSD) |
|---|---|---|---|
| run to run | 1.39 | 0.71 | 1.72 |
| day to day | 2.22 | 2.53 | 1.99 |
| column to column | 4.21 | 3.02 | 3.11 |

on the imprinted monolith decreased in the order of SMR, sulfamethazine, sulfadiazine, sulfathiazole, sulfamethoxazole, sulfisoxazole and sulfadimethoxine. The chromatograms also revealed that good separation of the seven SAs was obtained. In this study, the proper functional monomer supported three-dimensional cavities and provided strong application points for SMR. Although sulfamethazine was used as the template molecule and methacrylic acid and 4-vinylpyridine as co-functional monomers in He *et al.* [34] and Zheng *et al.*'s [35] research reports, the *I*-value of sulfamethazine (2.89) in this study was slightly below their values (3.04 and 3.60, respectively), and the *I*-value of SMR (3.34) was higher. (iv) The $R_s$ of the adjacent chromatographic peaks was 0.73, 0.98, 2.66, 2.34, 3.62, 1.67 in figure 9. The baseline separation of five SAs was achieved with the exception of sulfadimethoxine and sulfisoxazole on the imprinted monolith because of the different molecular recognition for SAs. The result indicates that the imprinted monolith could be used for the separation and determination of the different types of SAs. However, weak separation trends of SAs on non-imprinted monolith were found even under optimum chromatographic conditions, which was merely a process of simple adsorption and elution without specific selectivity.

## 3.5. Separation efficiency, reproducibility and stability of molecularly imprinted monolith

The separation efficiency is represented by the number of theoretical plates (*N*), which is calculated using the equation $N = 16(t_R/W)^2$, where $t_R$ is the retention time of each sulfonamide, and *W* is the width at the baseline between tangents drawn to inflection points for the peak. The average separation efficiency of 12 698 plates m$^{-1}$ was readily achieved. The separation efficiencies of the individual analytes were as follows: SMR, 16 200; sulfamethazine, 13 049; sulfadiazine, 12 909; sulfathiazole, 15 340; sulfamethoxazole, 11 971; sulfisoxazole 10 335; and sulfadimethoxine, 9082 plates m$^{-1}$.

Reproducibility and stability is a critical CEC parameter in the field of preparation and application of the imprinted monolith. In this study, the results of relative standard deviation (RSD) on the retention time and $R_s$ using SMR and sulfamethazine as testing samples are shown in table 3. The RSD for run to run, day to day, column to column was in range 0.71–4.21%. Moreover, the separation efficiency of the imprinted monolith only decreased by 3.60% over 200 runs in three consecutive months. These results demonstrated that the imprinted monolith had good reliability and application.

## 3.6. Method validation and application

The results of the linear calibration curves, recoveries, RSD and LOD are shown in table 4. The calibration curve was established by diluting a stock solution of SAs at different concentrations (0.1, 0.2, 0.5, 1.0, 2.0, 5.0 10.0, 20.0 mg kg$^{-1}$). As shown in table 4, a good linearity for SMR was achieved and the coefficient ($R^2$) was 0.998. The LOD was 0.040 mg kg$^{-1}$, which was estimated at a signal-to-noise ratio of three times. According to the Codex Alimentarius Commission (CAC/MRL 2–2011) and No. 235 Announcement of the Agriculture Ministry of China, the LOD was lower than the maximum residue limits (0.1 mg kg$^{-1}$) for SAs and suitable for the quantitative requirements of SMR in animal-derived food. To evaluate the applicability of the method, the recoveries for SMR were then investigated by using spiked blank fish and shrimp samples with SMR at three different levels (0.5, 2 and 20 mg kg$^{-1}$). Five sample replicates of each concentration were prepared and analysed in the optimum CEC condition. The results of the method recoveries ranged from 86.01% to 105.44%, with RSD values below 7.60%. The typical chromatograms of the blank fish sample before and after being spiked with SMR at 3 mg kg$^{-1}$ are displayed in figure 10. It can be observed that the retention ability of the imprinted monolith for trace-level SMR was stronger than for other impurities and successfully separated from the matrix compounds. Finally, the residue SMR was not detected in two commercial fish and one commercial shrimp samples using the developed method, which was in

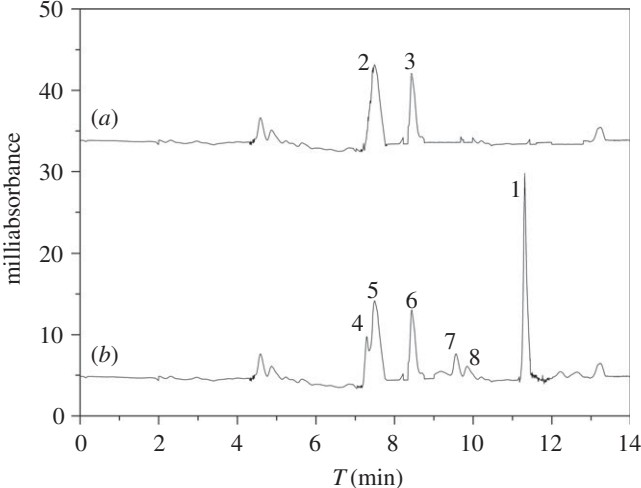

**Figure 10.** Electrochromatograms of blank fish sample (*a*) and spiked with SMR fish product (*b*). 1, SMR; 2, 3, 4, 5, 6, 7, 8—unknown impurity in the fish product.

**Table 4.** Recoveries, LOD and calibration curve of SMR.

| samples | spiked (mg kg$^{-1}$) | found (mg kg$^{-1}$) | recovery (%) | RSD (%) | calibration curve | LOD (mg kg$^{-1}$) |
|---|---|---|---|---|---|---|
| fish | 0.50 | 0.47 | 94.16 | 7.60 | $y^a = 768.13x^b + 1353.68$ | 0.040 |
| (n = 5) | 2.00 | 1.79 | 89.63 | 2.44 | | |
| | 20.00 | 2.11 | 105.44 | 4.02 | | |
| shrimps | 0.50 | 0.43 | 86.01 | 4.98 | | |
| (n = 5) | 2.00 | 0.202 | 101.00 | 3.19 | | |
| | 20.00 | 18.53 | 92.67 | 2.14 | | |

[a]The concentration (mg kg$^{-1}$) of SMR.
[b]The chromatographic peak area.

accordance with the detected results of high-performance liquid chromatography coupled with diode array detection (HPLC-DAD) analysis.

## 4. Conclusion

A new molecularly imprinted monolith containing carboxyl groups in the imprinted cavities was synthesized to perform separations of SA analogues using a single *in situ* preparation method. Complete separation of seven SAs was performed under the optimum CEC mode, in which the better $R_s$ of SMR and sulfamethazine separation achieved was 1.67. Moreover, the imprinted monolith not only simplified the preparation process but also showed strong selective recognition and good stability for the template, revealing that SMR could be successfully screened. Finally, the molecularly imprinted monolith coupled with the CEC method was established and successfully applied to the analysis of SMR in aquatic products without a solid phase extraction (SPE) clean-up procedure. The combination of molecularly imprinted technology with CEC could be applied in food analysis, and the imprinted monolith could present new perspectives for the development of a highly selective and efficient stationary phase.

Ethics. This experiment does not require an ethical assessment.
Data accessibility. The data supporting this article have been uploaded as a part of the electronic supplementary material.
Authors' contributions. S.Q. and P.W. were responsible for the examination of the thesis topic and the arrangement of the entire experimental project. L.G. was responsible for the revision of the article lang. L.S. and Y.L. were responsible for the operation of the entire experiment, the writing of the first draft of the paper and the analysis of the data. F.J. and S.H. were responsible for the use of the instruments.

Competing interests. We declare we have no competing interest.

Funding. We gratefully acknowledge the support of the basic special engineering general program of Heilongjiang Provincial Education Department (no. 135209215), 48th overseas returnees project from the Ministry of National Education, the overseas scholar scientific research project of Heilongjiang Provincial Education Department (no. 1254HQ012), Heilongjiang Province Science Foundation for Youths (no. QC2018073) and Fundamental Research Funds in Heilongjiang Provincial Universities (no. 135109201).

Acknowledgements. On behalf of the group, S.Q. would like to thank family and friends for their energetic support. S.Q. is also grateful to Ms Yang Xin for guiding our thesis writing, to Miss Yimin Tang for encouraging us and to Mr Qiang Dai, Mr Zhixiang Jin, Miss Xiaotong Lin and Miss Xue Li for helping us complete the experiment.

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
