## [Reviewer comments · Royal Society Open Science]

Review History

RSOS-190119.R0 (Original submission)

Review form: Reviewer 1

Is the manuscript scientifically sound in its present form?

Yes

Are the interpretations and conclusions justified by the results?

No

Is the language acceptable?

Yes

Is it clear how to access all supporting data?

No

Do you have any ethical concerns with this paper?

No

Have you any concerns about statistical analyses in this paper?

No

Recommendation?

Accept with minor revision (please list in comments)

Comments to the Author(s)

The present manuscript report on the synthesis of a molecularly imprinted phase and further coupling with capillary electrochromatography for the determination of sulfamerazine antibiotics. The paper is well-validated, topic is somewhat new and will be of great interest for the readership of Royal Society Open Science. Yet, some points (see below) still need to be addressed and I recommend publication after revision.

1. Please avoid the excessive use of abbreviations, as these made the paper difficult to read.
2. Authors should discuss on the possibility of mixing multiple MIPs phases specific to different types of sulfamerazine type antibiotics for multiplexed detection.
3. Maximum allowable levels of sulfamerazine or recommended guideline values are missing. Authors should include them and discuss on the adequacy of its LODs for the determination of such compounds.

Review form: Reviewer 2

Is the manuscript scientifically sound in its present form?

Yes

Are the interpretations and conclusions justified by the results?

Yes

Is the language acceptable?

No

Is it clear how to access all supporting data?

Not Applicable

Do you have any ethical concerns with this paper?

No

Have you any concerns about statistical analyses in this paper?

Yes

Recommendation?

Major revision is needed (please make suggestions in comments)

Comments to the Author(s)

The authors described a detailed report of using molecularly imprinted capillary electrochromatography to analyze sulfamerazine. It can possibly get accepted if the following issues are addressed.

1. Comparing to reference 32, the authors used a different functional monomer, what is the reason and advantages? Add a little comparison in the conclusion?
2. On page 2, line 10, the authors state sample requirement for CEC is three order of magnitudes lower than LC without any reference. I don't believe this is a sound statement. Detection limit is

highly depending on detector types. And it is also not the case that CEC peaks are three times narrower than LC peaks with same sample injection.

3. Page 2, line 55, "developed for the quantitative determination of SMR in aquatic products." Is a little misleading. The fact that all the samples doesn't have any SMR residues and there is no quantitative data for the spiked sample.

4. Page 8, after reading it three times, it is still not very clear to me how the method validation and application part was conducted. Was the calibration curve constructed with just standard solution or with spiked fish or shrimp sample? What is an aquatic sample? The description feels like standard solutions were used while the table 4 shows a different picture. Please clarify or rewrite that part.

5. The using of acronym should be more carefully, make sure they are defined properly before use (such as MIM) and an acronym name list would be helpful.

6. Page 4, line 52, "the I value (1.67) was lower." 1.87? possible typo?

7. Page 8, line 27, the equation for plate number calculation should be $16(tR/W)^2$

8. Page 8, line 37, reproductivity and repeatability is a challenge for in-situ CEC. The authors mentioned RSD of five repeated measurement for each column. How many columns were made? Variation between column? Any upward/downward trend in the reproducibility experiments (in other word, is there any degradation observed)? Limited of life time?

9. It is understandable there is some language challenge, however, some of sections are a little confusing. It is necessary to go through native-speakers for proofreading.

Decision letter (RSOS-190119.R0)

05-Apr-2019

Dear Ms Qin:

Title: Determination of sulfamerazine in aquatic products by molecularly imprinted capillary electrochromatography

Manuscript ID: RSOS-190119

The editor assigned to your manuscript has now received comments from reviewers. We would like you to revise your paper in accordance with the referee and Subject Editor suggestions which can be found below (not including confidential reports to the Editor). Please note this decision does not guarantee eventual acceptance.

Please submit your revised paper before 28-Apr-2019. Please note that the revision deadline will expire at 00.00am on this date. If we do not hear from you within this time then it will be assumed that the paper has been withdrawn. In exceptional circumstances, extensions may be possible if agreed with the Editorial Office in advance. We do not allow multiple rounds of revision so we urge you to make every effort to fully address all of the comments at this stage. If deemed necessary by the Editors, your manuscript will be sent back to one or more of the original reviewers for assessment. If the original reviewers are not available we may invite new reviewers.

To revise your manuscript, log into <http://mc.manuscriptcentral.com/rsos> and enter your Author Centre, where you will find your manuscript title listed under "Manuscripts with Decisions." Under "Actions," click on "Create a Revision." Your manuscript number has been

appended to denote a revision. Revise your manuscript and upload a new version through your Author Centre.

On behalf of the Subject Editor Professor Anthony Stace and the Associate Editor Professor John Moses.

RSC Associate Editor:
Comments to the Author:
(There are no comments.)

RSC Subject Editor:
Comments to the Author:
(There are no comments.)

Reviewers' Comments to Author:
Reviewer: 1

Comments to the Author(s)

The present manuscript report on the synthesis of a molecularly imprinted phase and further coupling with capillary electrochromatography for the determination of sulfamerazine antibiotics. The paper is well-validated, topic is somewhat new and will be of great interest for the readership of Royal Society Open Science. Yet, some points (see below) still need to be addressed and I recommend publication after revision.

1. Please avoid the excessive use of abbreviations, as these made the paper difficult to read.
2. Authors should discuss on the possibility of mixing multiple MIPs phases specific to different types of sulfamerazine type antibiotics for multiplexed detection.
3. Maximum allowable levels of sulfamerazine or recommended guideline values are missing. Authors should include them and discuss on the adequacy of its LODs for the determination of such compounds.

Reviewer: 2

Comments to the Author(s)

The authors described a detailed report of using molecularly imprinted capillary electrochromatography to analyze sulfamerazine. It can possibly get accepted if the following issues are addressed.

1. Comparing to reference 32, the authors used a different functional monomer, what is the reason and advantages? Add a little comparison in the conclusion?
2. On page 2, line 10, the authors state sample requirement for CEC is three order of magnitudes lower than LC without any reference. I don't believe this is a sound statement. Detection limit is highly depending on detector types. And it is also not the case that CEC peaks are three times narrower than LC peaks with same sample injection.
3. Page 2, line 55, "developed for the quantitative determination of SMR in aquatic products." Is a little misleading. The fact that all the samples doesn't have any SMR residues and there is no quantitative data for the spiked sample.
4. Page 8, after reading it three times, it is still not very clear to me how the method validation and application part was conducted. Was the calibration curve constructed with just standard solution or with spiked fish or shrimp sample? What is an aquatic sample? The description feels like standard solutions were used while the table 4 shows a different picture. Please clarify or rewrite that part.
5. The using of acronym should be more carefully, make sure they are defined properly before use (such as MIM) and an acronym name list would be helpful.
6. Page 4, line 52, "the I value (1.67) was lower." 1.87? possible typo?
7. Page 8, line 27, the equation for plate number calculation should be $16(tR/W)^2$
8. Page 8, line 37, reproductivity and repeatability is a challenge for in-situ CEC. The authors mentioned RSD of five repeated measurement for each column. How many columns were made? Variation between column? Any upward/downward trend in the reproducibility experiments (in other word, is there any degradation observed)? Limited of life time?
9. It is understandable there is some language challenge, however, some of sections are a little confusing. It is necessary to go through native-speakers for proofreading.

Author's Response to Decision Letter for (RSOS-190119.R0)

See Appendix A.

Decision letter (RSOS-190119.R1)

22-May-2019

Dear Ms Qin:

Title: Determination of sulfamerazine in aquatic products by molecularly imprinted capillary electrochromatography

Manuscript ID: RSOS-190119.R1

It is a pleasure to accept your manuscript in its current form for publication in Royal Society Open Science. The chemistry content of Royal Society Open Science is published in collaboration with the Royal Society of Chemistry.

On behalf of the Subject Editor Professor Anthony Stace and the Associate Editor Professor John Moses.

RSC Associate Editor
Comments to the Author:
(There are no comments.)

Reviewer(s)' Comments to Author:

Appendix A

Dear Reviewers,

Thank you very much for your review and comments on our manuscript (RSOS-190119)!

We have made all the possible corrections, and have revised the manuscript very carefully according to your kind suggestions.

If there are any other comments concerning this manuscript, please kindly let us know, and any comments you would give us will be greatly appreciated.

Thank you very much again!

Sincerely yours,

Shili Qin

Our replies are shown in italics.

Reviewer: 1

Thank you very much for your kind comments and good suggestions!

Based on the suggestions, I will answer the questions and modify the article.

Question 1: Please avoid the excessive use of abbreviations, as these made the paper difficult to read.

Reply 1: *Thanks for your query. As for the excessive use of abbreviations in this manuscript causing your inconvenience, we give our sincerely apologies here. Based on the suggestion, we have modified the most of abbreviations to full name and reserved some common and key words, such as SMR, CEC, LOD, SAs, CE, SPE, EGDMA and γ -MAPs.*

Question 2: Authors should discuss on the possibility of mixing multiple MIPs phases specific to different types of sulfamerazine type antibiotics for multiplexed detection.

Reply 2: *Thank you for your kind questions. Based on the suggestion, we have discussed a detailed description on page 8 and 9. The concrete content was showed in 4.3 “Imprinted effect evaluation of molecularly imprinted monolith” including the selective reason of seven sulfonamides. Meanwhile the data of Rs was supplemented and used to further evaluate the specific recognition of the imprinted monolith.*

“... .. Among them, sulfisoxazole, sulfamethazine and sulfathiazole belonged to short-acting SAs; sulfamerazine, sulfadiazine and sulfamethoxazole belonged to medium-acting SAs ; sulfadimethoxine belonged to long-acting SAs based on the pharmacological properties and clinical uses.^[33] The mixtures with similar structures, properties and representativeness could offer good explanations to investigate the

recognition mechanism. ... (4) The R_s of the adjacent chromatographic peaks was 0.73, 0.98, 2.66, 2.34, 3.62, 1.67 in Figure 9. The baseline separation of five SAs was achieved with the exception of sulfadimethoxine and sulfisoxazole on the imprinted monolith because of the different molecular recognition for SAs. The result indicated that the imprinted monolith could be used for the separation and determination of the different types of SAs. However the weak separation trends of SAs were on non-imprinted monolith even under the optimum chromatographic conditions, which was merely a process of the simple adsorption and elution without the specific selectivity.”

Question 3: Maximum allowable levels of sulfamerazine or recommended guideline values are missing. Authors should include them and discuss on the adequacy of its LODs for the determination of such compounds.

Reply 3: Thank you for your kind questions. It is well known that the LOD is important for the quantitative determination. The recommended guideline values have been supplemented in 4.4 section on page 9.

“... According to Codex Alimentarius Commission (CAC/MRL 2-2011) and NO. 235 Announcement of Agriculture Ministry of China, the LOD was lower the maximum residue limits (0.1 mg kg^{-1}) for SAs and suitable for the quantitative requirements of SMR in animal-derived food. ... ”

Reviewer: 2

Thank you very much for your kind comments and good suggestions!

Based on the suggestions, I will answer the questions and modify the article.

Question 1: Comparing to reference 32, the authors used a different functional monomer, what is the reason and advantages? Add a little comparison in the conclusion?

Reply 1: Thank you for your kind questions. Based on your suggestion, the main reason of itaconic acid as the functional monomer is the two carboxylate groups structure, which was not only as a proton donor and a hydrogen bonder forming the stable recognition

cavity but also as a generator of electroosmotic flow. The concrete content was showed in section 4.1, 4.3 ,4.4 on page 4, 7 and 8.

“... .. Itaconic acid with two carboxylate groups was used as a proton donor and a hydrogen bonder. In this study, both hydrogenbond (carboxylate of itaconic acid and amino of SMR) and electrostatic interactions (acid carboxylate of itaconic acid and basic pyrimidine of SMR) could be formed between the template and the functional monomer simultaneously. Compared with the molecularly imprinted polymer using methacrylic acid or 4-vinylpyridine as functional monomer, the imprinted polymer using itaconic acid with higher charged functionalities presented better separation efficiency and recognition ability. ^[28]... ..Moreover at this pH value the functional monomers partly dissociated which not only generated the EOF but also made the hydrogen bonding between the analytes and the functional groups stronger.In this study, the proper functional monomer supported three-dimensional cavities and provided strong application points for SMR. Although sulfamethazine was used as the template molecular and methacrylic acid and 4-vinylpyridine as co-functional monomers in He and Zheng ' s research reports, the I value of sulfamethazine (2.89) in this study was slightly below their values (3.04 and 3.60, respectively), and the I value of SMR (3.34) was higher.^[34,35] ”

Question 2: On page 2, line 10, the authors state sample requirement for CEC is three order of magnitudes lower than LC without any reference. I don't believe this is a sound statement. Detection limit is highly depending on detector types. And it is also not the case that CEC peaks are three times narrower than LC peaks with same sample injection.

Reply 2: *Thanks for your query. As for the vague station causing your inconvenience, we give our sincerely apologies here. In this section we want to state the small injection volume of CEC system because of hydrodynamic injection or electrokinetic injection, rather than the higher detection limit of CEC than LC. The small injection volume could save the precious biological samples. So we have modified this section and supplement the reference on page 2, line 8.*

“It is important to realize that CEC system do operate with small sample quantities (nanoliter amounts) and sample vials used for the introduction of the sample only require microliter quantities (typically $\geq 50 \mu\text{L}$), which is quite attractive for the precious biological samples.^[7]”

Question 3: Page 2, line 55, “developed for the quantitative determination of SMR in aquatic products.” Is a little misleading. The fact that all the samples doesn’t have any SMR residues and there is no quantitative data for the spiked sample.

Reply 3: Thank you for your kind questions. Based on your suggestion, we have listed the recovery data in Table 3 and modified the expression on page 2, line 55.

“... .. Finally, the molecularly imprinted monolith coupled with CEC method was established and applied to the determination of SMR in the aquatic products with adding standards to the blank, a satisfactory result was obtained.”

Tab.4 Recoveries, LOD and calibration curve of SMR

Samples	Spiked (mg kg-1)	Found (mg kg-1)	Recovery (%)	RSD(%)	Calibration curve	LOD (mg kg-1)
fish(n=5)	0.5 0	0.47	94.16	7.60	$y^a = 768.13x^b + 1353.68$	0.040
	2.00	1.79	89.63	2.44		
	20.00	2.11	105.44	4.02		
shrimps(n=5)	0.50	0.43	86.01	4.98		
	2.00	0.202	101.00	3.19		
	20.00	18.53	92.67	2.14		

Question 4: Page 8, after reading it three times, it is still not very clear to me how the method validation and application part was conducted. Was the calibration curve constructed with just standard solution or with spiked fish or shrimp sample? What is an aquatic sample? The description feels like standard solutions were used while the table 4 shows a different picture. Please clarify or rewrite that part.

Reply 4: *Thanks for your query. As for the vague station causing your inconvenience, we give our sincerely apologies here. In the experiment the calibration curve constructed with standard solution was accomplished according the national standard method. And this section have been rewritten in section 4.6 on page 9.*

“The results of the linear calibration curves, recoveries, RSD and LOD were shown in Table 3. The calibration curve was established by diluting stock solution of SAs at different concentrations (0.1, 0.2, 0.5, 1.0, 2.0, 5.0 10.0, 20.0 mg kg⁻¹). As shown in Table 4, a good linearity for SMR was achieved and the coefficient (R²) was 0.998. The LOD was 0.040 mg kg⁻¹, which was estimated at a the signal-to-noise ration of three times. According to Codex Alimentarius Commission (CAC/MRL 2-2011) and NO. 235 Announcement of Agriculture Ministry of China, the LOD was lower than the maximum residue limits (0.1 mg kg⁻¹) for SAs and suitable for the quantitative requirements of SMR in animal-derived food. To evaluate the applicability of the method, the recoveries for SMR were then investigated by using the spiked blank fish and shrimps samples with SMR at three different levels (0.5, 2 and 20 mg kg⁻¹). Five sample replicates of each concentration were prepared and analyzed in the optimum CEC condition. The results of the method recoveries ranged from 86.01-105.44%, with RSD values below 7.60%. The typical chromatograms of the blank fish sample before and after being spiked with SMR at 3 mg kg⁻¹ were displayed in Figure 10. It can be observed that the retention ability of the imprinted monolith for trace-level SMR was the strongest than for other impurities and successfully separated from the matrix compounds. Finally the residue SMR was not detected in two commercial fish and one commercial shrimps samples using the developed method, which was in accordance with the detected results of HPLC-DAD analysis.”

Question 5: The using of acronym should be more carefully, make sure they are defined properly before use (such as MIM) and an acronym name list would be helpful.

Reply 5: *Thanks for your query. As for the excessive use of abbreviations in this manuscript causing your inconvenience, we give our sincerely apologies here. Based on your and the other reviewer’s suggestions, we have modified the most of abbreviations to*

full name and reserved some common and key words, such as SMR, CEC, LOD, SAs, CE, SPE, EGDMA and γ -MAPs.

Question 6: Page 4, line 52, “the I value (1.67) was lower.” 1.87? possible typo?

Reply 6: *Thank you for your kind questions. Based on your question, the incorrect I value (1.67) have been modified to 1.87 according to the Table 1.*

Question 7: Page 8, line 27, the equation for plate number calculation should be $16(tR/W)^2$

Reply 7: *Thank you for your kind questions. Based on your question, the equation for plate number calculation have been modified.*

Question 8: Page 8, line 37, reproductivity and repeatability is a challenge for in-situ CEC. The authors mentioned RSD of five repeated measurement for each column. How many columns were made? Variation between column? Any upward/downward trend in the reproducibility experiments (in other word, is there any degradation observed)? Limited of life time?

Reply 8: *Thank you for your kind questions. The reproductivity and repeatability of the prepared imprinted monolith was indeed a challenge. The good results were obtained by optimizing the reaction condition and step-and-repeat operation. Based on your question, the data of reproductivity have been disposed and showed in section 4.5 on page 9.*

“... .. Reproducibility and stability is a critical CEC parameter in the field of preparation and application of the imprinted monolith. In this study, the results of RSD on the retention time and R_s using SMR and sulfamethazine as testing sample were showed in Table 4. As shown in Table 4, the RSD for run to run, day to day, column to column was in range from 0.71 to 4.21%. Moreover, the separation efficiency of the imprinted monolith only decreased 3.60% over 200 runs in three consecutive months. These results demonstrated that the imprinted monolith had good reliability and application. ”

Tab. 3 The reproducibility of the imprinted monolith

Type and numbers (n) of experiment	Rs (% RSD)	t _{SMR} (% RSD)	t _{sulfamethazine} (% RSD)
Run to run	1.39	0.71	1.72
Day to day	2.22	2.53	1.99
Column to column	4.21	3.02	3.11

Question 9: It is understandable there is some language challenge, however, some of sections are a little confusing. It is necessary to go through native-speakers for proofreading.

Reply 9: *Thank you for your kind questions. The manuscript has been checked carefully and the tense and spelling errors have been amended according to the reviewer's suggestions.*